# Host-defence caerin 1.1 and 1.9 peptides suppress glioblastoma U87 and U118 cell proliferation through the modulation of mitochondrial respiration and induce the downregulation of CHI3L1

Yichen Wang[1], Furong Zhong[1,2], Fengyun Xiao[1], Junjie Li[1,2], Xiaosong Liu[1,2,3], Guoying Ni[1,2,3], Tianfang Wang[4,5]*, Wei Zhang[1,3,6]*

1 The First Affiliated Hospital/Clinical Medical School, Guangdong Pharmaceutical University, Guangzhou, China, 2 Zhong'ao Biomedical Technology (Guangdong) Co., Ltd, Zhongshan, Guangdong, China, 3 Cancer Research Institute, First People's Hospital of Foshan, Foshan, Guangdong, China, 4 Centre for Bioinnovation, University of the Sunshine Coast, Maroochydore BC, QLD, Australia, 5 School of Science, Technology and Engineering, University of the Sunshine Coast, Maroochydore BC, QLD, Australia, 6 Guangdong Provincial Engineering and Technology Research Center of Stem Cell Therapy for Pituitary Disease, Guangzhou, China

* zw_gdpu@outlook.com (WZ); twang@usc.edu.au (TW)

## Abstract

Glioblastoma, the most aggressive form of brain cancer, poses a significant global health challenge with a considerable mortality rate. With the predicted increase in glioblastoma incidence, there is an urgent need for more effective treatment strategies. In this study, we explore the potential of caerin 1.1 and 1.9, host defence peptides derived from an Australian tree frog, in inhibiting glioblastoma U87 and U118 cell growth. Our findings demonstrate the inhibitory impact of caerin 1.1 and 1.9 on cell growth through CCK8 assays. Additionally, these peptides effectively curtail the migration of glioblastoma cells in a cell scratch assay, exhibiting varying inhibitory effects among different cell lines. Notably, the peptides hinder the $G_0$/S phase replication in both U87 and U118 cells, pointing to their impact on the cell cycle. Furthermore, caerin 1.1 and 1.9 show the ability to enter the cytoplasm of glioblastoma cells, influencing the morphology of mitochondria. Proteomics experiments reveal intriguing insights, with a decrease in CHI3L1 expression and an increase in PZP and JUNB expression after peptide treatment. These proteins play roles in cell energy metabolism and inflammatory response, suggesting a multifaceted impact on glioblastoma cells. In conclusion, our study underscores the substantial anticancer potential of caerin 1.1 and 1.9 against glioblastoma cells. These findings propose the peptides as promising candidates for further exploration in the realm of glioblastoma management, offering new avenues for developing effective treatment strategies.

**Data Availability Statement:** The mass spectrometry proteomics data have been deposited to the ProteomeXchange Consortium via the PRIDE partner repository with the dataset identifier PXD044941. The flow cytometric data of apoptosis experiment have been deposited to Flowrepository with the dataset link http://flowrepository.org/id/FR-FCM-Z764, while for the cell cycle experiment, the link to the data is http://flowrepository.org/id/FR-FCM-Z765.

**Funding:** This study was supported in part by the First Affiliated Hospital of Guangdong Pharmaceutical University, Deng Feng project of Foshan First People's Hospital (2019A008), Foshan municipal Government (2015AG1003), Guangdong Science and Technology Department (2016A020213001), National Science Foundation of Guangdong province (2020A1515010855), National Natural Science Foundation of China (31971355). The funders had no role in study design, data collection and analysis, decision to publish, or preparation of the manuscript.

**Competing interests:** The authors have declared that no competing interests exist.

## Introduction

Glioblastoma, also known as glioblastoma multiforme (GBM), is a highly aggressive and malignant type of brain tumour that originates from glial cells, which can cause a variety of physical symptoms, such as headaches, seizures, speaking disorders, and impaired motor skills, as well as cognitive impairment [1, 2]. As a tumour originated in the central nervous system (CNS), it poses significant challenges due to its fast infiltrative growth and limited response to conventional therapies [3]. Despite the use of surgery, chemotherapy, and radiotherapy, which can be costly, their efficacy remains limited, with a median survival period of around 14.2 months [4]. New therapeutic approaches are imperative to enhance the prognosis and treatment of GBM [5].

Although clinical approval for drugs is yet to be granted, therapeutic cancer vaccines and adoptive cell metastasis therapies are burgeoning fields in basic and clinical research on brain tumours. Significant strides have been made in active vaccine immunotherapy and the adoptive therapy of immune effector cells [6]. The emerging vaccine treatment paradigm zeroes in on targeting tumour-specific antigens, incorporating approaches involving glioma stem cells, isocitrate dehydrogenase mutations, and heat shock proteins into diverse clinical trials [7]. However, immunotherapy grapples with several challenges in GBM treatment. The blood-brain barrier (BBB) poses a barrier to the entry of immune cells and therapeutic agents, restricting their access to glioblastoma cells [8]. Tumour heterogeneity, characterised by diverse cell types, complicates effective targeting [9], and the immunosuppressive tumour microenvironment (TME), featuring regulatory T cells and suppressor cells, impedes immune responses [10], thereby diminishing the efficacy of checkpoint inhibitors. The identification of specific antigens for immunotherapy is also challenging due to the absence of well-defined targets [11]. The rapid growth of glioblastomas outpaces immune responses, and low immune cell infiltration further impedes therapy, while glioblastoma cells can develop resistance mechanisms [12]. Overcoming these challenges necessitates innovative strategies and a more profound understanding of the glioblastoma microenvironment.

Natural peptides offer advantages over traditional drugs, such as high purity, controlled quality, and adjustable structure [13]. Their remarkable therapeutic efficacy and specificity position them as promising agents for diverse diseases, including cancers [14]. Among these peptides, caerin peptides, sourced from the skin glands of Australian tree frogs (Litoria), have garnered attention. Caerin 1.1 exhibits potent anti-tumour effects on various cancer cells *in vitro*, while caerin 1.9 demonstrates significant antibacterial properties [15–19]. Recent studies highlight the synergistic impact of combining caerin 1.1 and caerin 1.9, inducing apoptosis in cervical cancer cells and triggering the TNF-α signalling-mediated pathway [15]. Moreover, this combination inhibits tumour growth *in vivo* by activating IFNAR-1 and STAT1 signalling pathways in tumour-infiltrating macrophages, shifting macrophage polarisation from M2 to M1 types [20, 21]. This activation subsequently stimulates tumour-infiltrating T cells, NK cells, and dendritic cells. Notably, caerin 1.1 and caerin 1.9 enhance the efficacy of anti-PD-1 and therapeutic vaccine-based immunotherapy in a tumour model [21]. However, the potential therapeutic effect of caerin peptides on GBM remains uninvestigated.

The primary objective of this study is to examine the impact of caerin 1.1 and 1.9 peptides on the growth of glioblastoma cells, specifically U87 and U118 cells. Furthermore, the study involves the utilisation of quantitative proteomic analysis to delve into the molecular mechanisms that underlie the biological effects of caerin peptides on glioblastoma cells. This endeavour holds the potential to offer insights applicable to *in vivo* assessments and offer a direction for the advancement of innovative therapies for GBM.

## Materials and methods

### Cell line

The glioma cell lines (U87 and U118) were procured from the National Collection of Authenticated Cell Cultures, China. U87 MG cells were cultured in complete medium supplemented with 10% heat inactivated FBS, 90% DMEM medium and 1% Penicillin-Streptomycin. These cells were cultured at 37°C with 5% $CO_2$. Similarly, U118 MG cells were cultured in complete medium supplemented with 10% heat inactivated FBS, 90% DMEM medium and 1% Penicillin-Streptomycin. These cells were also cultured at 37°C with 5% $CO_2$. U87 and U118 cell lines have different genetic mutations, including TP53, PTEN, EGFR and CDKN2A.

### Peptide synthesis

Caerin 1.1 (named as F1, GLLSVLGSVAKHVLPHVVPVIAEHL-NH$_2$), caerin 1.9 (named as F3, GLFGVLGSIAKHVLPHVVPVIAEKL-NH$_2$) , and a control peptide P3 without cytotoxic properties towards various cancer cells (GTELPSPPSVWFEAEFK-OH) were synthesised by Shanghai Qiangyao Biological Technology Co., Ltd, China. The peptides' purity was determined to be 99.47% (F1), 99.55% (F3), and 99.29% (P3), respectively, through reverse-phase high-performance liquid chromatography (HPLC). The caerin peptides along with the P3 peptides were stored at 4°C until their use.

### CCK8 assay

Cell proliferation was assessed using a CCK8 assay (Glpbio, GK10001) as per the manufacturer's instructions. Before the addition of the peptides, U87 and U118 cells were individually cultivated in flat-bottomed 96-well plates for 18 h to ensure successful cell adhesion. Various concentrations (ranging from 0 to 10 μg/ml) of F1/F3 (molar ratio 1:1) were introduced to $1 \times 10^4$ U87 cells, followed by a 6 h incubation at 37°C with 5% $CO_2$. Similarly, different concentrations (ranging from 0 to 20 μg/ml) of F1/F3 (molar ratio 1:1) were added to $1 \times 10^4$ U118 cells, which were then incubated for 6 h at 37°C with 5% $CO_2$. After this, 10 microliters of CCK8 Reagent were added and the cells were cultured for an additional 2 h. The outcomes were examined using a microtiter plate reader (BioTek, USA) at 570 nm, as per the manufacturer's protocol. The IC50 value for each caerin peptide was computed employing a method described elsewhere.

### Wound-healing assay

U87 and U118 cells were cultured in a flat bottomed 12 well plate for 18 h, with $1 \times 10^5$ cells seeded per well. The culture medium was then removed from each well and the cells were washed with PBS. Sterile pipette tips were used to create controlled scratches in each well, and then the cells were wash again with PBS. Subsequently, images were taken using an inverted microscope (Leica, Germany) to record the initial state of the wound. The 12 well plates were divided into an untreated group and an experimental group. The experimental group was added 1 mL of culture medium containing F1/F3 (the concentration of F1/F3 is 5 μg/mL, the molar ratio 1:1), while the untreated group was added an equal amount of culture medium without F1 or F3. Then, the culture plates were placed in the incubator for different durations (6 h, 12 h, and 24 h) and the images were captured again using an inverted microscope (Leica, Germany). The width of the scratch wound was measured as an indicator of cell migration or closure by comparing the initial and final images.

## Fluorescence microscopy

U87 and U118 cells were cultured separately in flat bottomed 12 well plates. The 12 wells were divided into three groups. The first group was used as an untreated group and only added to the culture medium. The second group was used as an experimental group and added to the culture medium containing 5 μg/mL F1/F3 (the molar ratio 1:1) labelled by FITC (the isothiocyano group of FITC is linked together with the amino group of the peptide after reaction). The third group was used as a control group and added to the culture medium containing 5 μg/mL P3 peptide marked by FITC, allowing all groups of cells to react for 6 h in the incubator (37˚C with 5% $CO_2$). Afterwards, the supernatant was removed from each well and 50 μL of DAPI (Beyotime Biotech, China) was added to react at room temperature for 5 min. DAPI dye was extracted from each well and cleaned with PBS (Gibco, USA). Finally, 100 μL of culture medium was added to each well and the distribution of FITC labelled F1/F3 in glioblastoma cells was observed under a fluorescence microscope.

## Cell apoptosis assay

The assessment of cell apoptosis was conducted using a Cell Apoptosis Kit (Biosharp, BL107A) as per the manufacturer's guidelines. Briefly, U87 and U118 cells were cultured for 18 h in two flat bottomed 6-well plates, with $5 \times 10^5$ cells seeded per well set. For the untreated group, 3 wells were selected from each well plate and only culture medium was added. For the experimental group, 3 wells were chosen and culture media containing F1/F3 were added. The control group comprised 3 wells with culture medium containing the P3 peptide. The remaining 3 wells were solely provided with culture medium for future voltage regulation. Subsequently, all cells were treated with trypsin (Biosharp, BL107A) in ice water, followed by washing with precooled PBS (Gibco, C10010500BT). AnnexinV-FITC and Propidium iodide were introduced for flow cytometry analysis using FACSARia II (BD Biosciences, San Jose, CA, USA). The outcomes were assessed utilising FlowJo v10.0 software (Tree Star Inc., Ashland, OR, US).

## Cell cycle experiment

Cell cycle analysis was carried out using a Cell Cycle Kit (C1052, Beyotime Biotech, China,) following the manufacturer's instructions. Briefly, U87 and U118 cells were cultured for 18 h in two flat bottomed 6-well plates, with $5 \times 10^5$ cells seeded per well set. For the untreated group, 3 wells were selected from each well plate, and only culture medium was added. For the experimental group, 3 wells were chosen, and culture medium containing F1/F3 was added. Following the guidelines of the cell cycle kit, all cells were dissolved and gathered using trypsin (Biosharp, BL107A), then washed with precooled PBS (Gibco, C10010500BT). Subsequently, the cells were fixed with 70% ethanol that had been precooled in an ice bath. After that, they were stained with a prepared Propidium iodide staining solution and eventually examined using flow cytometry (FACSAria II; BD Biosciences, San Jose, CA, USA). The results were evaluated using FlowJo v10.0 software (Tree Star Inc., Ashland, OR, USA).

## Electron microscope photography

U87 and U118 cells were cultured separately in flat-bottomed 12-well plates. Each plate was divided into four groups. Culture media containing 5 μg/mL F1 and F3 (the molar ratio 1:1) were added to each group based on different time points (0, 1, 6, and 12 h). The cells in each group were then placed in a culture incubator (37˚C, 5% $CO_2$) for the specified time duration. Subsequently, the cells were harvested using trypsin (Biosharp, China), washed with precooled PBS (Gibco, USA), and fixed with 2.5% glutaraldehyde (Biosharp, China). The fixed

samples were placed on an ultra-thin slicer (Leica uc7, Leica, Germany) for slicing, and then the organelle structure in different samples was observed under an electron microscope (Hitachi7500, HITACHI, Japan).

## Protein extraction, TMT-10plex labelling and high pH reversed-phase fractionation

Under identical conditions, both U87 and U118 cells, treated with F1/F3 or left untreated, were collected and rapidly frozen in liquid nitrogen for subsequent protein extraction. For each treatment and control, biological triplicates were gathered. The collected cell samples were thoroughly homogenised in SDT buffer (containing 4% SDS, 100 mM Tris-HCl pH 7.6, and 0.1M DTT) while maintained at 4˚C. The total protein content was determined using the Pierce BCA protein assay on a NanoDrop 2000 instrument (Thermo Fisher Scientific, Bremen, Germany). Samples containing 200 μg of total protein were subjected to trypsin digestion using the filter-aided proteome preparation method, as described elsewhere [22]. The resulting tryptic peptides were subsequently desalted using C18 Cartridges (Empore™ SPE Cartridges C18, Sigma), followed by lyophilization. The quantification of peptides was carried out using the NanoDrop 2000.

For further analysis, samples consisting of 100 μg of peptides obtained from uninfected, control, and treated cells were labelled with TMT10-plex reagents according to the manufacturer's instructions (Thermo Scientific, Waltham, MA, USA). Subsequently, the labelled samples were mixed and subjected to fractionation using a PierceTM high pH Reversed-Phase Peptide Fractionation Kit (Thermo Fisher Scientific, IL, USA), as directed by the manufacturer. All fractions were then lyophilized using a SpeedVac and reconstituted in 12 μL of 0.1% formic acid (FA) for LC-MS/MS analysis.

## Liquid chromatography tandem mass spectrometry analyses

Each sample solution (10 μL) was subjected to analysis using a two-dimensional EASY-nLC1000 system coupled with a Q Exactive Hybrid Quadrupole-Orbitrap Mass Spectrometer from Thermo Scientific. The analysis method employed was similar to that previously described in detail [23, 24]. In brief, the samples were introduced into the sample loading column (Thermo Scientific Acclaim PepMap100, 100 μm × 2 cm, nanoViper C18) and subsequently fractionated through the analytical column (Thermo Scientific EASY column, 10 cm, ID75μm, 3μm, C18-A2).

Operating in positive ion mode, the mass spectrometer acquired MS data using a data-dependent approach, starting with a survey scan ranging from 300 to 1800 m/z, followed by HCD fragmentation. The automatic gain control (AGC) target was set to 3E6, and the maximum injection time was set at 10 ms. A dynamic exclusion duration of 40.0 s was implemented. The survey scans were obtained at a resolution of 70,000 at m/z 200, while the resolution for HCD spectra was set to 17,500 at m/z 200. An isolation width of 2 m/z was used. The normalized collision energy was set to 30 eV, and the underfill ratio was defined as 0.1%. The mass spectrometer was operated with peptide recognition mode enabled. The proteomics data from the mass spectrometry analysis have been deposited in the ProteomeXchange Consortium via the PRIDE [25] partner repository with the dataset identifier PXD044941.

## Protein identification and quantification

The MS/MS data was searched against Homo sapiens (76,413 sequences, downloaded on Dec 12, 2014) database for protein identification using Mascot2.2 (Matrix Science, London, UK) and Proteome Discoverer1.4 software (Thermo Fisher Scientific, Waltham, MA, USA) with

the following search settings: enzyme trypsin; two missed cleavage sites; precursor mass tolerance 20 ppm; fragment mass tolerance 0.1 Da; fixed modifications: Carbamidomethyl (C), TMT 10plex (N-term), TMT10 plex (K); variable modifications: oxidation (M), TMT 10plex (Y). The results of the search were further submitted to generate the final report using a cut-off of 1% FDR on peptide levels and only unique peptides were used for protein quantitation. All peptide ratios were normalised by the median protein ratio, and the median protein ratio was 1 after the normalisation. The protein showing a fold change $\geq 1.2$ (upregulation $\geq 1.2$ or downregulation $\leq 0.83$) compared to the untreated group and the $P$-value $< 0.05$ were considered significantly regulated by the treatment and included in further analysis.

## Protein-protein interaction (PPI) analysis

Interactions among significantly regulated proteins were predicted using STRING [26]. All resources were selected to generate the network and 'confidence' was used as the meaning of network edges and the required interaction score of 0.400 was selected for all PPI, to highlight the confident interactions. Neither the 1st nor 2nd shell of the PPI was included in this study. Protein without any interaction with other proteins was excluded from displaying in the network.

## Gene ontology, KEGG pathway and GSEA analysis

The enrichment of biological processes and KEGG pathways [27] was assessed for the treatments in comparison to the untreated and control groups. The genes associated with the proteins exhibiting differential expression across the three groups were subjected to analysis using Gene Set Enrichment Analysis (GSEA) with a significance threshold of $P$-value $< 0.05$. This analysis was performed using GSEA version 4.1.0 [28, 29].

## Western blotting

Different concentrations (range from 0 to 10 µg/mL) caerin peptide added to $1 \times 10^4$ U87 cells were incubated at 37˚C and 5% $CO_2$ for 6 h. Similarly, different concentrations (ranging from 0 to 20 µg/mL) caerin peptide added to $1 \times 10^4$ U118 cells at 37˚C and 5% $CO_2$ for 6 h. The concentration range was decided based on their respective IC50 values. Cells were lysed in lysis buffer and electrophoresed in 10% SDS-PAGE gels, and then probed with specific antibodies purchased from Affinity Biosciences LTD, which included CHI3L1 (DF7223), PZP (AB122718), and JUNB (AF6198). PageRuler™ Prestained NIR Protein Ladder (Cat no. 22635, Thermo Fisher Scientific) was used to indicate the molecular weights of the bands.

## Glycolysis/oxidative phosphorylation assay

The Glycolysis/OXPHOS Assay Kit (G270, DOJINDO Laboratories, Kumamoto, Japan) was utilised to measure changes in cell energy metabolism before and after F1/F3 treatment following the manufacturer's instructions. In brief, U87 cells ($1 \times 10^4$) and U118 cells ($1 \times 10^4$) were seeded into a 96-well plate and incubate for 18 h at 37˚C in a 5% $CO_2$ incubator. Subsequently, the culture medium was removed, and the cells were treated with culture medium containing 5 µg/mL F1/F3 (molar ratio 1:1). Then, the culture medium was replaced and 100 µL oligomycin (+), or 2-dG (+) peptide-free culture medium was respectively added to the wells and incubated at 37˚C in a 5% $CO_2$ incubator for 5 h. 20 µL of supernatant was retrieved from each well for the lactate assay. Then, the remaining cells were subjected to the ATP assay in a 96-well plate. Finally, the absorbance values obtained from the lactate assay and the luminescence values of the ATP assay were compared, respectively.

### TCGA glioblastoma data analysis and survival analysis

The glioblastoma (GBM) datasets were obtained from The Cancer Genome Atlas (TCGA) database (https://portal.gdc.cancer.gov/repository), which contains 153 tumour samples and 5 normal samples. The datasets were transformed into Transcripts Per Million (TPM) values. The survival analyses were performed by the Cox proportional hazard model provided that the proportional hazard assumption was met based on weighted residuals using TIMER2.0 [30]. Hazard ratio was estimated relative to the lowest-risk group and assessed by a two-sided Wald test, $P$-value < 0.05 was significant. The split expression percentage of patients was set to 20%, and the survival time between 0 and 60 months was presented.

### Statistical analysis

Statistical analysis in this study, unless otherwise specified, employed an unpaired Student's t-test using GraphPad Prism 8 software. All experimental data underwent analysis, and graphs were generated using the same software. The determination of statistically significant means was based on a probability level of 0.05.

## Results

### F1/F3 inhibit the proliferation and migration of glioblastoma cells

In our study, we subjected U87 and U118 cells to various concentrations of F1/F3 for a 6 h period. The results obtained from the CCK8 assay underscored the considerable inhibitory effect of F1/F3 on U87 cell proliferation, as evidenced by an IC50 value of 5.018 μg/mL (**Fig 1A**). Similarly, the CCK8 experiment conducted on U118 cells indicated a substantial reduction in their proliferation, with an IC50 value of 11.180 μg/mL (**Fig 1B**). Despite this commonality, the CCK8 results elucidate a significant variance in their sensitivity to the F1/F3 treatment against U87 and U118 cells.

The migration of the glioblastoma cells was assessed by a scratch assay following the treatment of F11/F3 at different time points (0, 6, 12, and 24 h). The appearance of U87 and U118 cells differed; the degree of scratch healing was time-dependent for both the treatment and control groups. The scratch was nearly healed at 24 h in both control groups, while certain areas of the scratch were still visible in the treatment groups (**Fig 1C**). The healing rates were significantly reduced in the treated groups compared to the controls (**Fig 1D and 1E**). Notably, U118 exhibited a slower rate of scratch healing in the presence of F1/F3 compared to that of U87.

### F1/F3 induced apoptosis of U87 and U118 cells

The addition of F1/F3 at 5 μg/mL to the culture of either U87 or U118 cells induced U87 and U118 cell apoptosis. The percentage of apoptotic cells in treated U87 was 38.2% (**Fig 2A**), whereas it was 10.4% and 10.8% in the untreated and P3 groups, respectively. For U118 cells, 9.44% displayed apoptosis, which was still higher than those in the control groups (**Fig 2B**). Additionally, the induction of apoptosis by F1/F3 in U87 and U118 cells was highly significant compared to the controls, with a more pronounced effect observed in U87 cells (**Fig 2C**).

### F1/F3 entered the cytoplasm of glioblastoma cells and affected subcellular organelles

Next, we utilised FITC-labelled F1/F3 to track their cellular location during the treatment, with PBS and FITC-labelled P3 treated cells serving as the untreated and control groups,

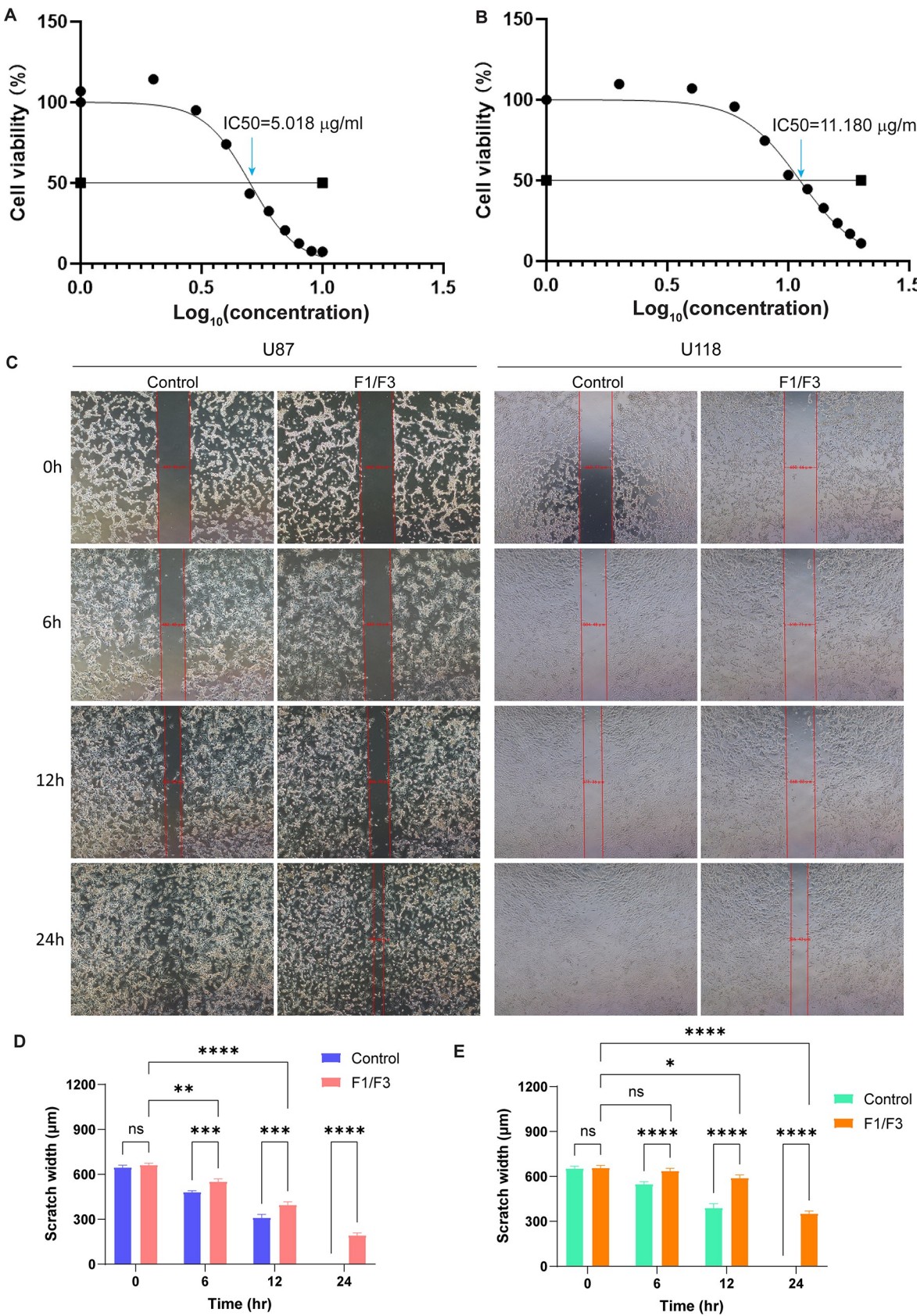

**Fig 1. CCK8 assay of the effects of F1/F3 against the proliferation of U87 and U118 cells.** (**A**) F1/F3 exerted a substantial inhibitory effect on the growth of U87 cells, with an IC50 value of 5.018 μg/mL. (**B**) F1/F3 significantly suppressed the proliferation of U118 cells, yielding an IC50 value of 11.18 μg/mL. Microscopic photos of the scratches in U87 (left) and U118 (right) cells recorded at 0, 6, 12, and 24 h in the control and treatment groups (F1/F3 at a concentration of 5 μg/mL, with a molar ratio 1:1), respectively (**C**). Comparison of the healing degree of U87 (**D**) and U118 (**E**) cells between the control and the treatment groups in the scratch experiment at different time. Data are representative of at least 3 independent experiments and are presented as the mean ± SD. ns: non-significant, *: $P$-value < 0.05, **: $P$-value < 0.01, ***: $P$-value < 0.001 and ****: $P$-value < 0.0001.

respectively. After 6 h of treatment, we observed green fluorescence inside the cells of the treatment groups, which was not observed in the control groups (**Fig 3A and 3B**). The intensity of green fluorescence was more prominent in U87 relative to that of U118 cells. Furthermore, we found that FITC-labelled F1/F3 were primarily distributed in the cytoplasm of the cells, with almost no presence in the nucleus of U87 or U118 cells, respectively (**Fig 3C**).

Subsequently, we utilised electron microscopy to scrutinize the cellular microstructures of glioma cells treated with F1/F3 (**Fig 4**). The results unveiled the profound impact of F1/F3 on various organelles, including mitochondria, endoplasmic reticulum, and the Golgi apparatus, of U87 (**Fig 4A**) and U118 cells (**Fig 4B**). Notably, the size of mitochondria increased with the duration of treatment, accompanied by a shift in their shapes towards a more spherical morphology. Furthermore, the degree of mitochondrial structural alterations was more pronounced in U87 cells, indicating a substantial impact on cellular morphology and organelle integrity.

## F1/F3 induced a reduction in mitochondrial transmembrane potential and alters the cell cycle of glioblastoma cells

Given the observed entry of F1/F3 into the cytoplasm of glioblastoma cells and its notable impact on mitochondrial swelling, we delved deeper into this effect. We employed JC-1 dye to label the mitochondria of glioblastoma cells and assessed the mitochondrial membrane potential after the addition of F1/F3 for 6 hours. Comparative analysis with a control group revealed a substantial decrease in mitochondrial membrane potential in both U87 (**Fig 5A**) and U118 (**Fig 5B**) cells treated with F1/F3, which exhibited significant high intensity of green fluorescence, compared to their untreated counterparts. Notably, this reduction was more pronounced in U87 cells. Additionally, F1/F3 influenced the timing of mitosis initiation, shortening the interval between DNA replication completion and mitosis onset in both cell lines. Simultaneously, it extended the time from DNA replication completion to the initiation of mitosis. U87 cells exhibited a more significant degree of change compared to U118 cells (**S1 Fig**).

## F1/F3 activated inflammatory response in U87 cells

TMT-10plex labelled quantitative proteomic analysis was employed to compare the protein profiles of untreated and treated cells, to shed light into the molecular mechanism underlying the anti-proliferative effects of F1/F3 on both U87 and U118 cells. Notably, a substantial number of DEPs (1,979) were detected in F1/F3 treated U118 cells when compared to U87 cells, and this pattern was consistent when comparing untreated groups (2,193 DEPs) (**Fig 6A**, **S1** and **S2** **Tables**). Interestingly, there was an overlap of 1,627 DEPs between the profiles of untreated and treated cell groups. Furthermore, the treatments (U87F and U118F) led to the regulation of 148 and 172 proteins, respectively. Simultaneously, this caused 285 and 249 DEPs that were originally present in the untreated group comparison to become insignificant. Regarding the hallmark pathways analysed, it was observed that three pathways related to cell proliferation and growth exhibited higher activation levels in U87 cells according to GSEA.

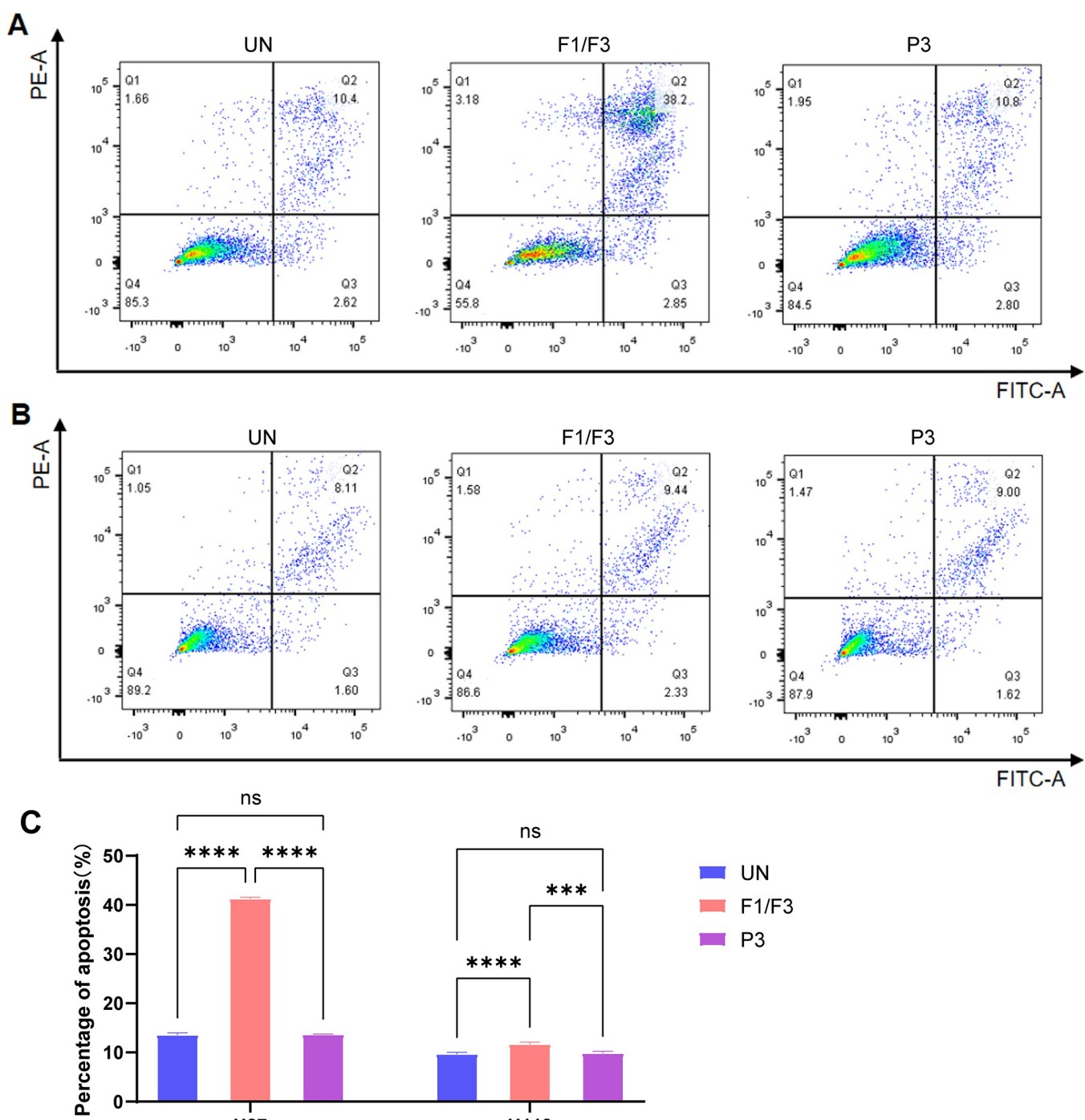

**Fig 2. F1/F3 induced the apoptosis of U87 and U118 cells.** Flow cytometry images compares the apoptosis degree of U87 (**A**) and U118 (**B**) cells in the untreated, F1/F3 and P3 groups. (**C**) Percentage of apoptotic cells in response to different treatments in U87 and U118 cells relative to the untreated groups, respectively. Data are representative of at least 3 independent experiments and are presented as the mean ± SD. ns: non-significant, ***: $P$-value < 0.001, ****: $P$-value < 0.0001.

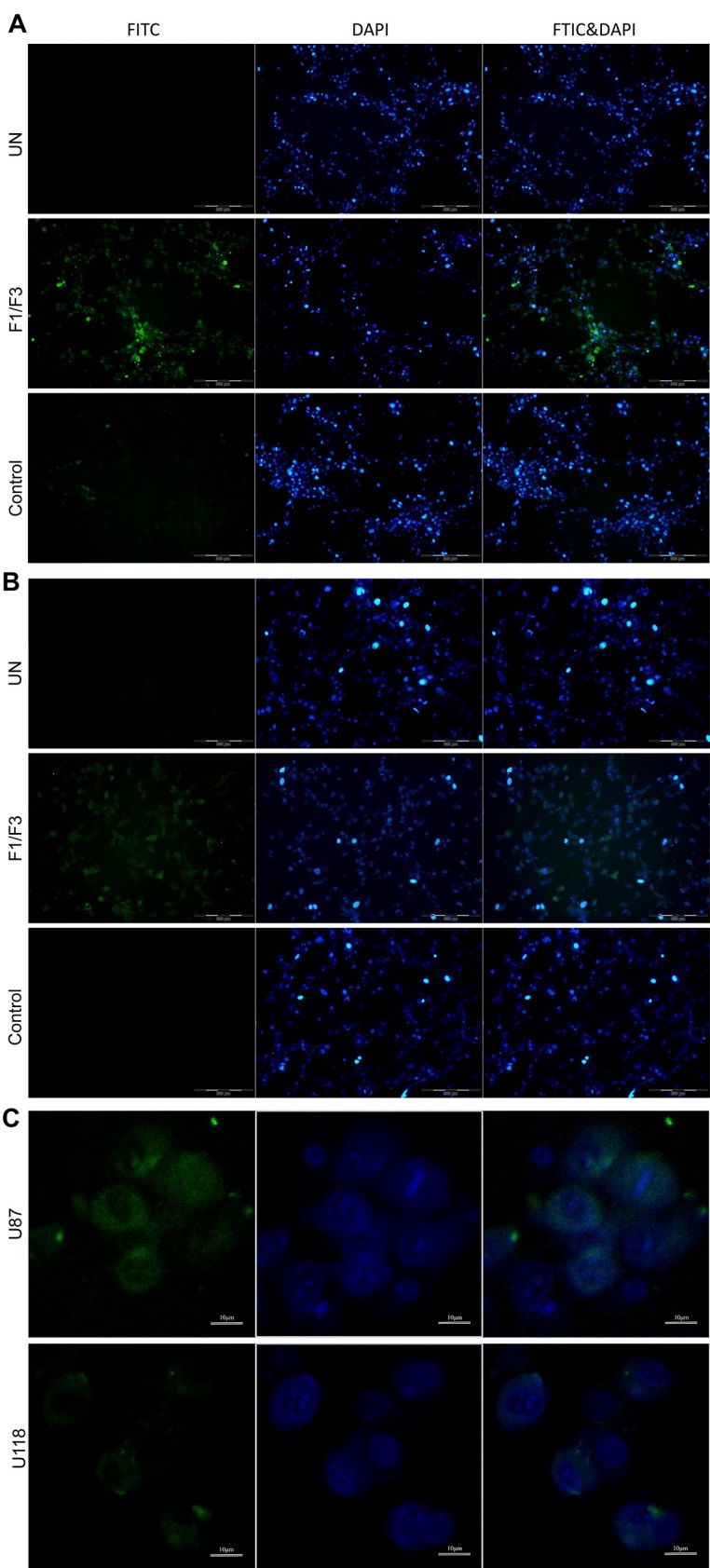

**Fig 3. Fluorescence microscopy images of glioma cells treated with PBS (untreated), FITC-labelled F1/F3 and FITC-labelled P3 (control) after 6 hours of treatment.** The concentration of F1/F3 is 5 μg/mL, and the molar ratio of F1 to F3 is 1:1. Fluorescence microscopic images of U87 (**A**) and U118 (**B**) cells in the untreated, F1/F3 and control groups with FITC and DAPI labels, as well as the merged images. (**C**) The distribution of FITC-labelled F1/F3 in the cytoplasm of U87 and U118 cells, respectively. In these images, blue corresponds to the nucleus stained with the DAPI probe, while green corresponds to FITC.

These pathways include "mTORC1 signalling," "apical junction," and "glycolysis" (**S2A Fig**). Conversely, U118 cells showed enrichment in two inflammation-relevant pathways, "INFγ response" and "IL6/JAK/STAT3 signalling," as well as four pathways associated with tissue growth support. In terms of gene ontology, numerous metabolic processes were found to be activated in U87 cells, while processes related to immune response were more prominent in U118 cells (**S2B Fig**). When examining molecular functions, it was noted that more proteins in U87 cells were significantly associated with "binding," whereas activities related to channel function and peptidase regulation were more representative in U118 cells (**S2C Fig**). Furthermore, a higher number of extracellular proteins were detected in U118 cells (**S2D Fig**). These findings provide insights into the distinctive molecular characteristics and functions of these two cell lines.

A total of 42 differentially expressed proteins (DEPs) were observed to be upregulated in the treatment containing F1/F3, while 22 DEPs were downregulated in U87 cells (**Fig 6A** and **S1 Table**). When comparing treated U118 cells with untreated groups, only three proteins (NEU3, RPLP1, and TENT4B) were identified as DEPs. One of the highly upregulated DEPs in treated U87 cells was PZP, also known as pregnancy zone protein or α2-HS glycoprotein (AHSG), with the highest fold change (FC) of 1.935 (**Fig 6B**). Several other significantly upregulated proteins were also related to immune responses, including ALB, SERPINE1, A2M, JUNB, and THBS1. The protein-protein interactions among the DEPs were predicted, revealing significant interactions among many upregulated DEPs. In this network, ALB emerged as the central node with the highest degree of interaction, followed by FN1 (**Fig 6C**). Notably, there were interactions between ALB and the downregulated proteins SCD and HMGA2. Additionally, interactions were identified among HLA-DPA1, HLA-DRA, and CTSK. These interactions suggest potential molecular pathways and associations that may contribute to the observed effects of F1/F3 on U87 cells.

The enrichment analysis of gene ontology in the U87F group revealed several biologically relevant processes related to immune responses that were enriched following the treatment. These processes included "response to stress," "inflammatory response," and "acute inflammatory response" (**Fig 6D** and **S3 Fig**). In terms of molecular functions (MF), the enrichment analysis highlighted "fibrinogen binding" as the most highly enriched function, followed by "MHC class II receptor activity." Additionally, three other molecular functions, namely "ferroxidase activity," "chemokine binding," and "complement component C1q binding," were also enriched (**S3A Fig**). Many of the proteins identified were associated with extracellular regions and the extracellular matrix (**S3B Fig**). Notably, the term "Fibrinogen complex" represented the most enriched cellular component (CC), even though the number of proteins falling under this term was relatively low. Interestingly, the analysis predicted the enrichment of proteins sourced from platelet components, including "platelet alpha granule lumen" and "platelet alpha granule." This prediction was supported by the presence of proteins such as ALB, IL6, FGA, AGT, SAA1, and APOA1 (as detailed in **S1 Table**). Additionally, the term "blood microparticle" also showed enrichment, suggesting a similarity between these cellular components and F1/F3 treated U87 cells.

The GSEA revealed the activation of several stress-relevant hallmark pathways in both treatment groups compared to the untreated groups. These pathways included "TNFα signaling via

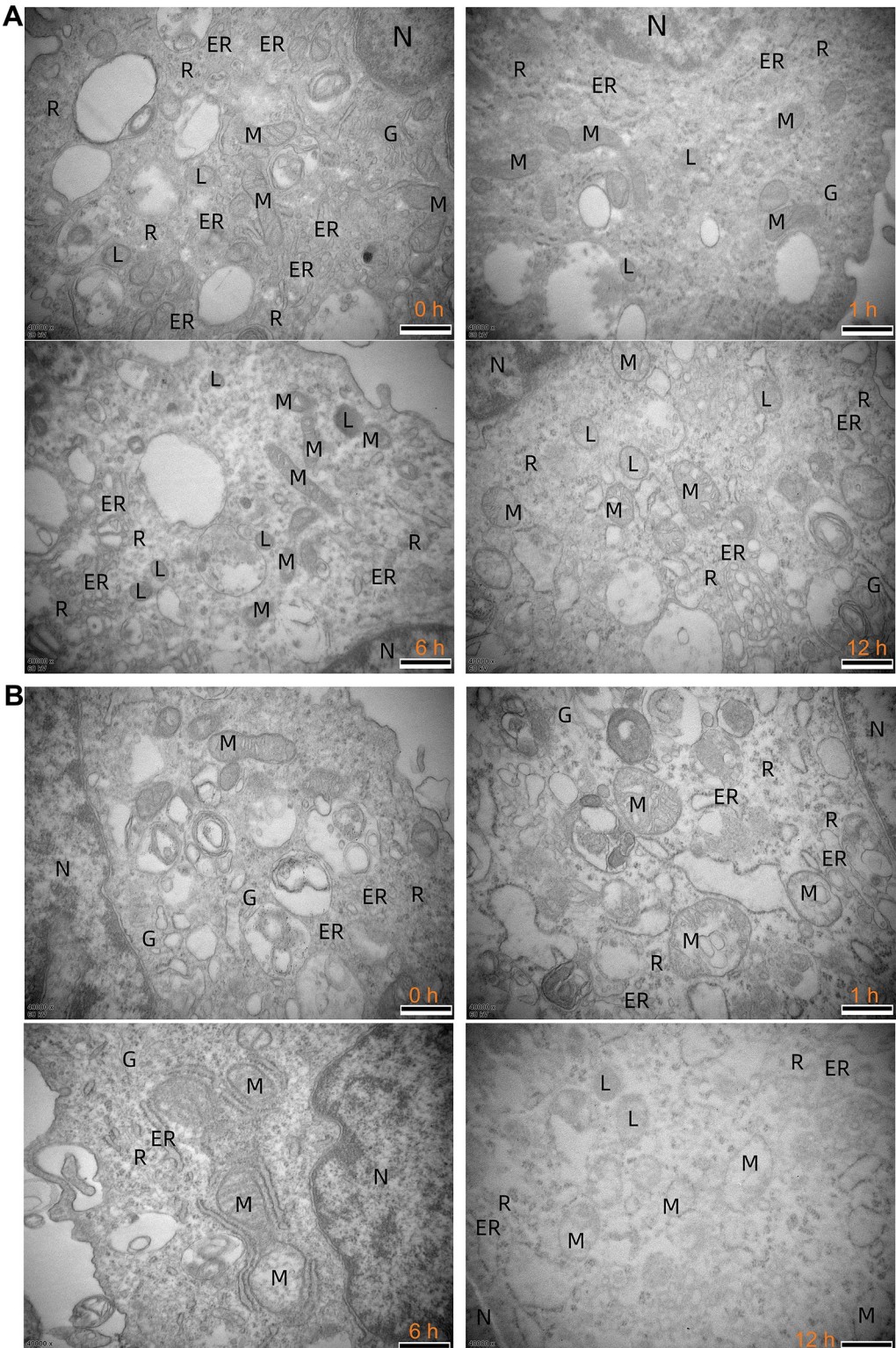

**Fig 4. Electron microscopy images of F1/F3 treating glioblastoma cells.** The magnification of the electron microscope is 40,000x, and the concentration of F1/F3 is 5 μg/mL with a molar ratio 1:1. Selected images of U87 (**A**) and U118 (**B**) cells treated by F1/F3 for 0, 1, 6, and 12 h. ER: endoplasmic reticulum; G: Golgi apparatus; L: lysosome; M: mitochondrion; N: nucleus; R: ribosome. The scale bar is 500 nm.

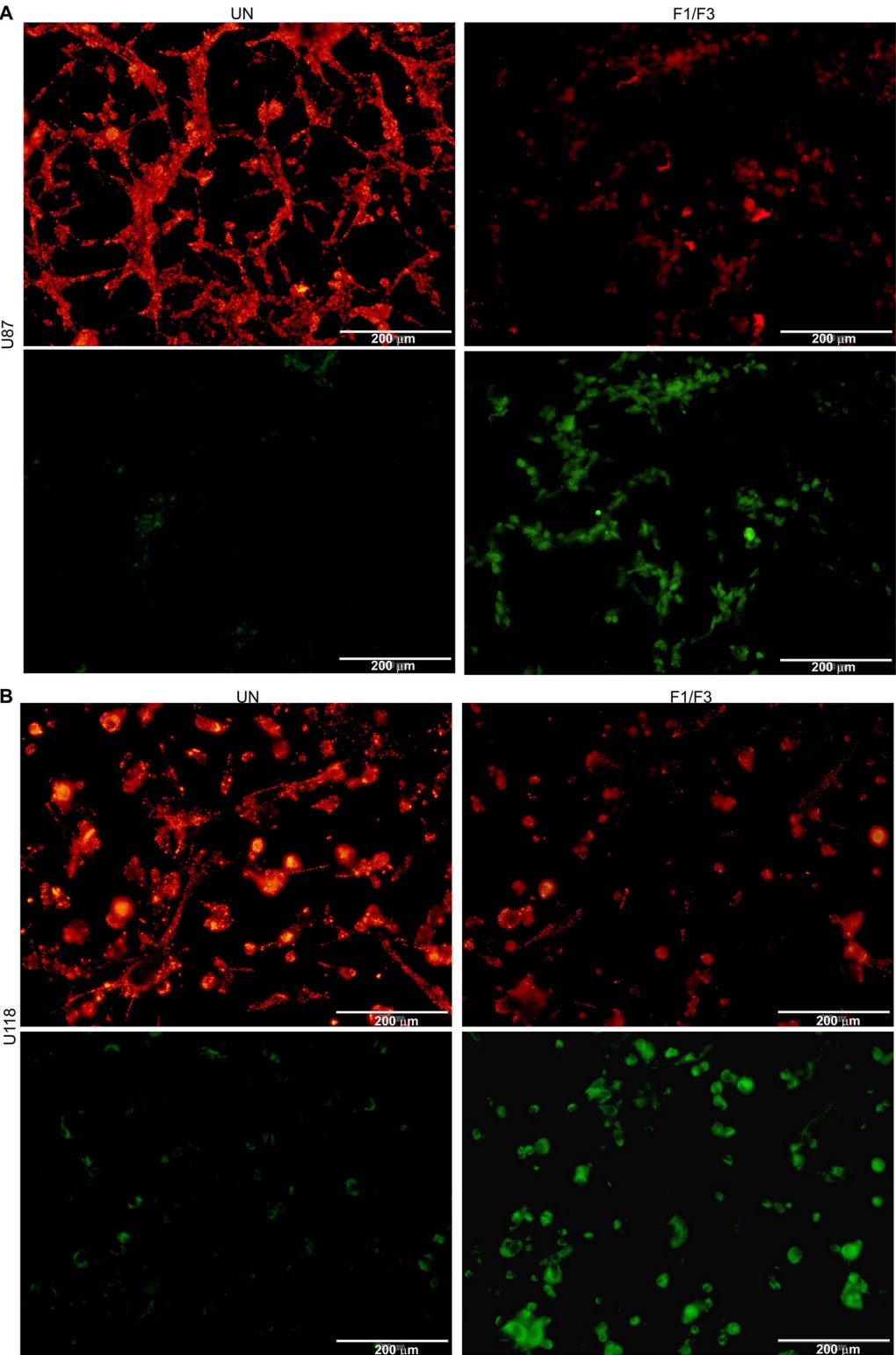

**Fig 5. The comparison of mitochondrial membrane potential in glioblastoma cells with and without F1/F3 treatment.** (**A**) Fluorescence microscopy images show the changes in mitochondrial membrane potential in untreated (left two images) and treated (right two images) U87 cells. (**B**) Fluorescence microscopy images show the changes in mitochondrial membrane potential in untreated (left two images) and treated (right two images) U87 cells. In the images, red indicates that JC-1 is in a polymer state, signifying healthy mitochondria, while green indicates that JC-1 is in a monomeric state, indicating a decrease in mitochondrial transmembrane potential.

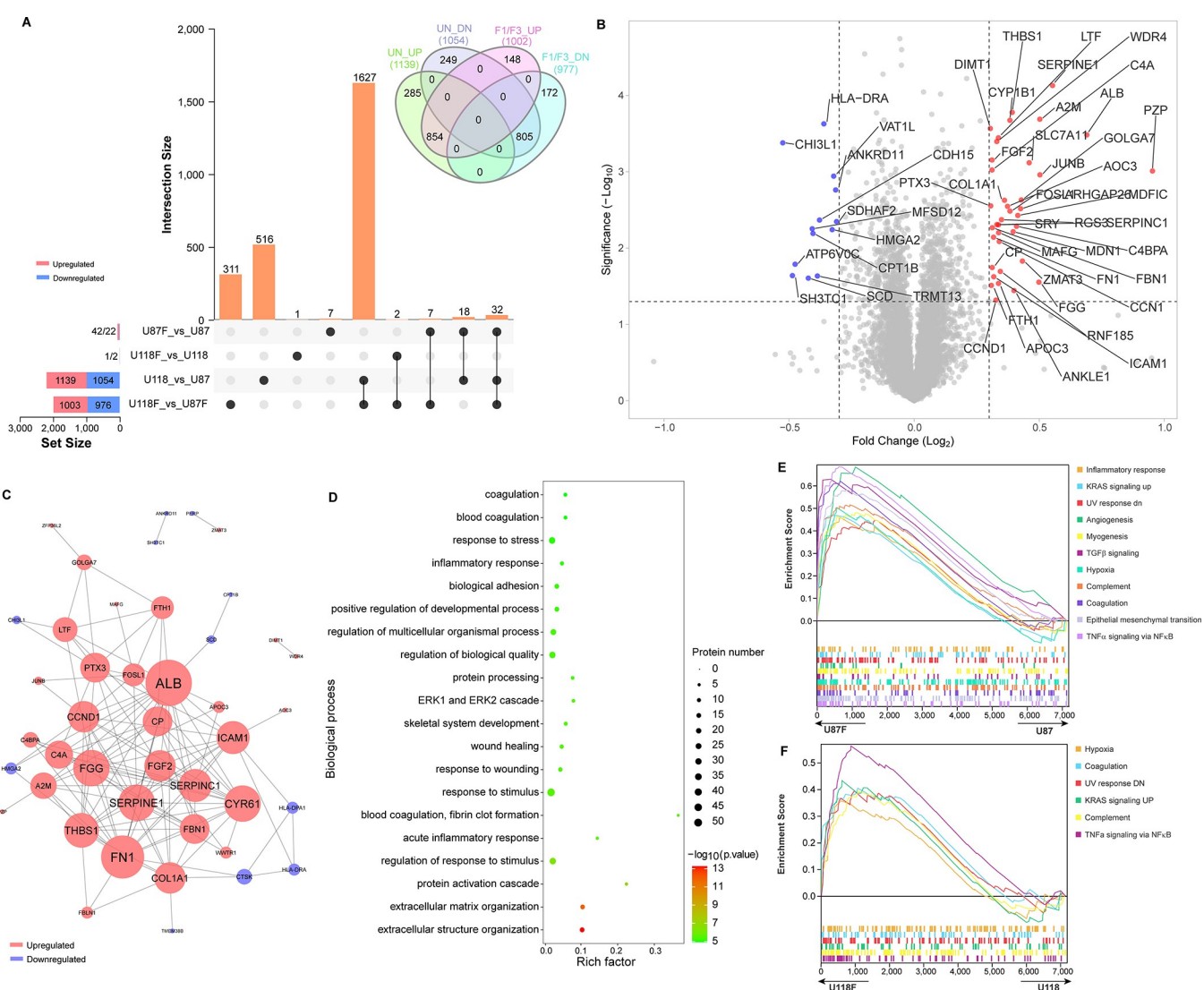

**Fig 6. Quantitative proteomic analysis of glioblastoma cells with F1/F3 treatment.** (**A**) The UpSet graph provides a comparison of the DEPs identified between different groups: U87, U118, U87F, and U118F. The Venn graph in (**A**) compares the DEPs up- and downregulated in the U87 relative to U118 group (UN), as well as the U87F relative to U118F group (F1/F3). (**B**) The Volcano graph represents the DEPs that are upregulated and downregulated in U87 cells due to the treatment, as compared to the untreated U87 group. (**C**) Protein-protein interaction between DEPs in the U87F relative to the U87 groups. (**D**) Enrichment analysis of biological processes in the U87F with respect to the U87 groups. GSEA reveals the hallmark pathways activated in the U87F (**E**) and U118F (**F**) concerning U87 and U118 groups, respectively. The data are representative of three independent experiments. (See **S1** and **S2 Tables** for detailed proteomic analysis results).

NFκB," "complement," "hypoxia," and "UV response dn." However, the activation level of these pathways was more prominent in the U87F group (**Fig 6E and 6F**). Of particular note, two immune response-associated pathways, namely "inflammatory response" and "TGFβ signalling," exhibited high enrichment specifically in the U87F group compared to the U87 group. Additionally, several pathways related to tissue development and wound healing, such as "epithelial mesenchymal transition," "angiogenesis," and "myogenesis," also showed high enrichment in the U87F group. This suggests that these pathways may be associated with cellular response to the greater antiproliferative effect of F1/F3 against U87 cells when compared to U118 cells (**Fig 6E**).

## F1/F3 significantly regulated the expression of CHI3L1, JUNB and PZP, as well as altered the metabolism in U87 cells

To validate the findings from proteomic analysis, we performed Western Blotting on three proteins—CHI3L1, JUNB, and PZP—that exhibited significant expression changes post F1/F3 treatment. At 5 µg/mL, F1/F3 notably downregulated CHI3L1 content compared to the U87 group (**Fig 7A**), with original gel images showed in **S1 Raw images** and data processing recorded in **S3 Table**. Subsequently, U87 cells were treated with F1/F3 concentrations ranging from 0 (control group) to 10 µg/mL, revealing a concentration-dependent decrease in CHI3L1 content (**Fig 7B**). In the case of U118 cells, a significant decreased content of CHI3L1 at 5 µg/mL F1/F3 was observed (**Fig 7C**), yet the dose dependence was not detected with increasing F1/F3 concentrations of 4, 8, 12, 16, and 20 µg/mL (**Fig 7D**). Conversely, the expression of *JUNB* and *PZP* showed an upregulation with higher concentrations of F1/F3 in treated U87 cells (**S4A** and **4B Fig**, **S1 Raw images**, and **S3 Table**). Exploring glioblastoma multiforme

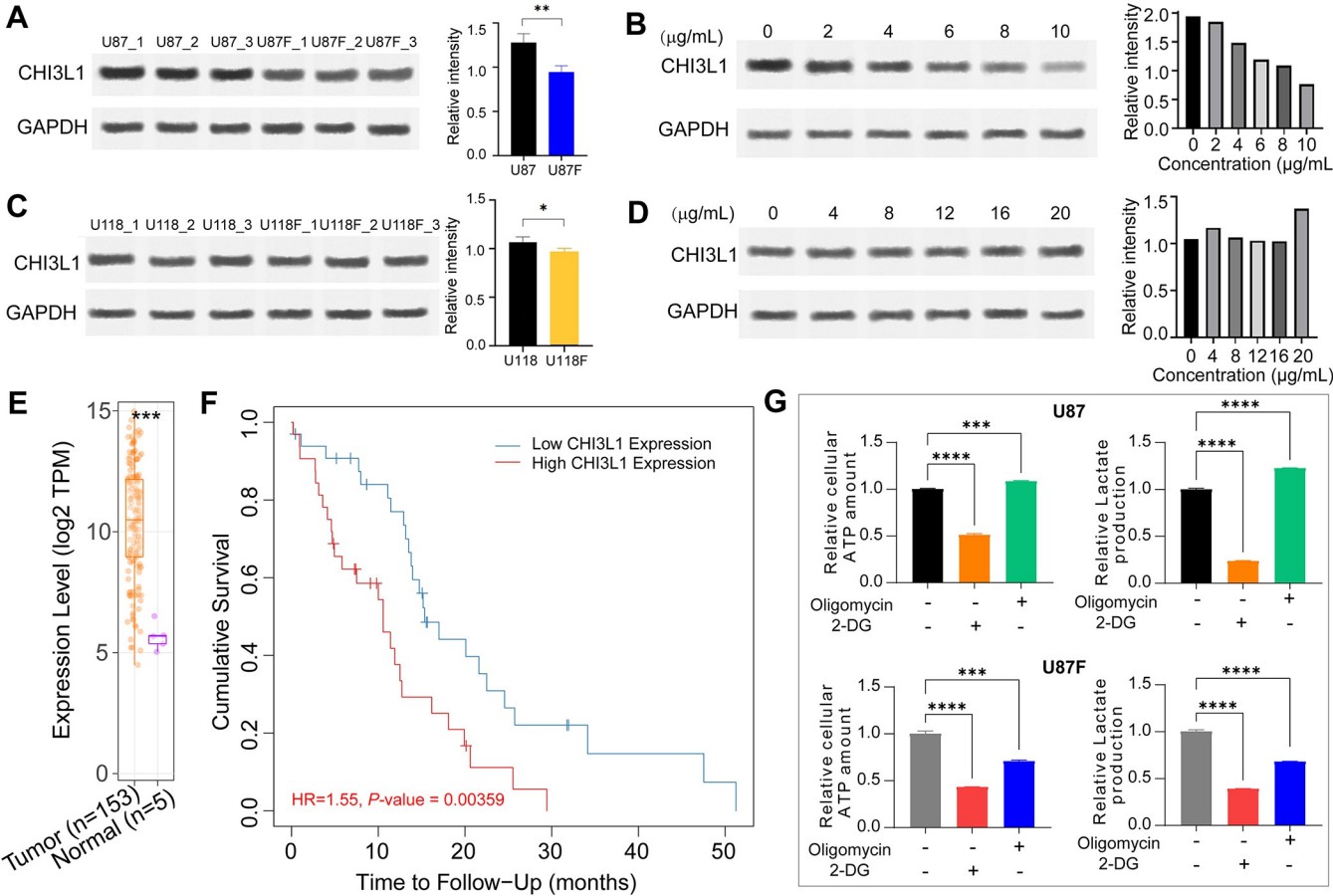

**Fig 7. F1/F3 treatment inhibited the expression of CHI3L1 and shifted the energy metabolism of U87 cells.** At 5 µg/mL, F1/F3 downregulated the content of CHI3L1 significantly (**A**) and induced a concentration-dependent response (**B**) in U87 cells. At 5 µg/mL, F1/F3 significantly downregulated the content of CHI3L1 (**C**) in U118 cells, and the concentration dependence displayed in (**D**). (**E**) The TCGA glioblastoma multiforme database revealed a notable upregulation of *CHI3L1* in tumours compared to normal tissues. (**F**) Low *CHI3L1* expression was positively correlated with the survival time of glioblastoma multiforme patients, showing statistical significance. (**G**) Evaluation of glycolysis/OXPHOS levels in the U87 (top) and U87F (bottom) groups demonstrated that F1/F3 treatment led to a shift from glycolysis to OXPHOS in U87 cells, as indicated by changes in ATP levels and lactate production. The inhibition of glycolysis or OXPHOS pathways was facilitated using 2-DG or oligomycin, respectively. Data are representative of at least 3 independent experiments and are presented as the mean ± SD. *: *P*-value < 0.05, **: *P*-value < 0.01, ***: *P*-value < 0.001, and ****: *P*-value < 0.0001.(see **S3 Table** for analysis results of western-blotting and TCGA data, **S1 Raw images** for original western-blotting images, and **S4 Table** for glycolysis/OXPHOS test data).

data on TCGA for *CHI3L1* expression revealed a significant upregulation in tumours compared to normal tissues (**Fig 7E**), and low expression correlated with longer patient survival at five years (*P*-value of 0.00359, **Fig 7F**).

Next, we assessed the levels of glycolysis and oxidative phosphorylation in U87 and U118 cells treated with F1/F3 at a concentration of 5 μg/mL. In the U87 group, ATP levels were approximately 0.5 and 1.1 with the addition of 2-DG or oligomycin, respectively, compared to the control; relative lactate production changed to 0.25 and 1.2 with the addition of 2-DG and oligomycin, respectively (**Fig 7G**). These findings suggested a high dependence on glycolysis for energy metabolism in U87 cells. In contrast, the U87F group exhibited a slight decrease in ATP levels by 0.1 when the glycolytic pathway was inhibited but a significant reduction to about 0.7 with OXPHOS suppression; lactate relative production markedly decreased to approximately 0.7 with OXPHOS inhibition, indicating a shift from glycolysis to OXPHOS in U87 cells treated with F1/F3. Conversely, there was minimal change in the energy metabolism of U118 cells treated with the same concentration of F1/F3, where glycolysis remained the dominant pathway consistently (**S4C Fig**).

## Discussion

In this study, we investigated the impact of a mixture comprising caerin 1.1 (F1) and caerin 1.9 (F3) at a molar ratio of 1:1 on the proliferation of two glioblastoma cell lines, namely U87 and U118 cells, through *in vitro* experiments. Our findings revealed that F1/F3 exhibited a stronger anti-proliferative effect on U87 cells, with an IC50 value of 5.018 μg/mL, compared to an IC50 of 11.180 μg/ml observed for U118 cells. This discrepancy in potency corresponded to varying levels of apoptosis observed in the two cell lines with the treatment of F1/F3. Moreover, we observed the penetration of F1/F3 into the cells, leading to interactions with various organelles within the cells. Notably, treatment with F1/F3 induced mitochondrial swelling over time, and alterations in mitochondrial membrane potential were prominently detected, especially in U87 cells. Furthermore, through quantitative proteomic analysis, we identified many DEPs upregulated in U87 cells upon treatment with F1/F3, which were associated with the significant activation of inflammatory response and the TNFα signalling pathway in U87 cells.

Proteomic comparison of the protein profiles of these two cell lines was limited. Our proteomic analysis has now identified a substantial number of DEPs between the untreated U87 and U118 groups, as well as between the U87F and U118F groups after treatment with F1/F3. The observed differences in protein profiles reflect the significantly distinct phenotypes exhibited by U87 and U118 cells. U87 cells appeared to be more actively engaged in various metabolic processes that support cell proliferation, making them more sensitive to any energy metabolism alteration (**S2 Fig**). Conversely, U118 cells exhibited a greater presence of acute phase response signalling. The higher level of activation in pathways such as mTORC1 signalling, hypoxia, and glycolysis in U87 cells aligns with the reprogramming of tumour metabolism towards aerobic glycolysis, which is often associated with increased invasiveness [31]. These findings suggest that U87 cells may possess a more invasive phenotype compared to U118 cells. Interestingly, recent reports have highlighted differences between U87 and U118 cells in terms of gene expression, particularly in genes related to extracellular vesicles, such as cMET and CD44. U87 cells were found to exhibit significantly higher expression levels of both genes [31]. Additionally, differences in invasiveness within hydrogels were observed, with U87 cells displaying more proliferation [32]. Furthermore, a comparison of their secretomes revealed distinct signatures for U87 cells, including the presence of ADAM9, ADAM10, cathepsin B, cathepsin L1, osteopontin, neuropilin-1, semaphorin-7A, suprabasin, and CHI3L1 [33]. Remarkably, in our observation, procathepsin L, semaphorin-7A, and

suprabasin were also identified as DEPs that were upregulated in U87 cells relative to U118 cells. The differential enrichment of cellular components among the identified proteins suggests that the intercellular environment is of greater importance to U87 cells, while the extracellular space is relatively more crucial for U118 cells. These molecular characteristics may indeed be correlated with the varying sensitivity displayed by U87 and U118 cells to F1/F3 treatment.

CHI3L1 (Chitinase-3-like protein 1) was significantly downregulated in the U87F group (**Fig 6**). CHI3L1 is a member of the glycoside hydrolase family 18 [34], is known to play crucial roles in oxidative injury, apoptosis, pyroptosis, extracellular matrix regulation, Th1/Th2 inflammatory balance, and parenchymal scarring [35–37]. It has been implicated in promoting glioblastoma progression, particularly through its association with angiogenesis [38–40]. High levels of CHI3L1 are linked to more aggressive tumour behaviour and increased invasiveness [41]. Additionally, CHI3L1 has immunosuppressive effects within the glioblastoma microenvironment and has been associated with resistance to certain therapeutic interventions, presenting a significant challenge in glioblastoma management [42, 43]. Elevated CHI3L1 levels have been correlated with poor prognosis, shorter overall survival, and disease progression in glioblastoma patients [44, 45]. A recent study also reported increased expression of CHI3L1 and HLA-DRA in reactive astrocytes derived from glioblastoma, with HLA-DRA being inhibited in the treated group [46]. In this study, we demonstrated that F1/F3 significantly reduced the content of CHI3L1 in U87 cells in a dose-dependent manner (**Fig 7**). Interestingly, CHI3L1 was not detected as a DEPs in the U118F group by the proteomic analysis, yet the westernblotting analysis revealed a significant decrease with 5 μg/mL of F1/F3, which might be related to the differing responses of U87 and U118 cells to F1/F3. Additionally, a lack of dose dependence was observed for CHI3L1 downregulation in U118 cells. Considering the proteomic analysis identified that the untreated U118 cells were more enriched with immune response relevant biological processes compared to untreated U87 cells (**S2B Fig**), while F1/F3 treatments have been found to activate interferon-γ signalling pathway in the TME [15, 20, 21], potentially counteracting the anti-proliferative effects of F1/F3, particularly at concentrations above a certain threshold. This phenomenon has been observed in immunotherapies that regulate various signalling pathways. For instance, tamoxifen demonstrates this effect in estrogen receptor signalling [47], while several MEK inhibitors exhibit biphasic effects in the MAPK/ERK pathway [48–50].

Pregnancy zone protein (PZP), a member of the proteinase inhibitor I39 (-2-macroglobulin) family, has gained attention for its role in immune modulation [51–53]. Recent studies indicate its correlation with the clinical stage of lung adenocarcinoma, acting as an independent unfavourable prognostic factor [54]. High PZP expression increased CD86+ M1 macrophages and decreased CD206+ M2 macrophages in lung adenocarcinoma. Downregulation of PZP in hepatocellular carcinoma is linked to poor prognosis, correlating with elevated levels of macrophages and neutrophils [55]. In addition, it was found that resting memory CD4 T cells were increased in high PZP expression group. In breast cancer, low PZP expression is associated with tumour progression and reduced survival. Genetic ablation of PZP promotes tumorigenesis [56], and its hypermethylation and low expression in hepatocellular carcinoma suggest a role in inhibiting proliferation, invasion, and migration [57]. In the current study, F1/F3 treatment led to the highest upregulation of PZP in U87 cells (**Fig 6**), suggesting a potential link between PZP, anti-proliferative activity, and the activated immune response signalling induced by F1/F3, such as inflammatory response. which warrants further verification.

Fluorescence microscopy confirmed the penetration of F1/F3 into both U87 and U118 cells (**Fig 3**), which was also observed in F1/F3 treated HeLa cells with a time-dependent manner [15]. In this study, a significant enrichment of TNFα signalling via NFκB was detected in both

cell lines. Electron microscopy revealed the direct impact of F1/F3 on various organelles, especially mitochondria, aligning with alterations in mitochondrial membrane potential following treatment (**Fig 4**). Mitochondria play a critical role in various metabolic processes, including those that were comparatively activated in U87 cells by F1/F3. The observed changes in mitochondrial morphology and swelling may be linked to treatment-induced stress. It thus suggested a potential shift in energy metabolism from glycolysis to OXPHOS, which could potentially slow down cell growth, as OXPHOS is a less proliferative form of metabolism compared to glycolysis [58, 59]. The shift was confirmed in U87 cells treated with 5 μg/mL F1/F3 yet was less observable in U118 cells (**Fig 7**). The reduction in mitochondrial membrane potential might be a counter effect of the cells in response to this shift, to compensate for reduced mitochondrial function and to move away from OXPHOS.

This metabolic shift was more pronounced in U87 cells treated with 5 μg/mL F1/F3 than in U118 cells, potentially contributing to hindered $G_0$/S phase replication, particularly prominent in U87 cells (**S1 Fig**). Moreover, the treatment may activate cellular stress responses, including apoptosis, reflected in the upregulation of DEPs associated with the "regulation of epithelial cell apoptotic process," such as *SERPINE1*, *FGG*, *THBS1*, *ICAM1*, and *FGB*. The significant stimulation of TNFα signalling in the U87F group implied a cellular response to the stress, akin to observations in F1/F3-treated HeLa cells [15] and TC-1 cells [16]. The significant modulation of this pathway was also reported in *in vivo* TC-1 tumour mouse models treated with F1/F3-containing drug candidates through topical application and intertumoral injection [23, 24], which also showed that F1/F3 can induce the reprogramming of immunosuppressive M2-like to inflammatory M1-like macrophages in the TME. This observation could be linked to the activation of inflammatory response signalling in the treated U87 cells. F1/F3's ability to alter the immunosuppressive microenvironment to be more immune response active could potentially address a challenge in adopting immunotherapy for treating GBM [60]. Additionally, F1/F3 markedly recruited more activated immune cells, such as CD8[+] T cells, *MHCII*[hi] macrophages, and NK cells, to the tumour sites of TC-1 mouse models [17, 24]. This recruitment might possibly enhance the infiltration of immune cells in GBM, which generally exhibit low levels compared with other tumours [61].

While F1/F3 demonstrated inhibitory effects on glioblastoma cell proliferation, a significant challenge lies ahead. The molecular weights of F1/F3 may hinder their easy penetration of the blood-brain barrier (BBB) if considered as drug candidates. The BBB is a complex structure, and factors such as lipophilicity, charge, and specific transport mechanisms can influence a substance's ability to cross. Although F1/F3 show promising cell penetration abilities with U87 and U118, their impact on other normal brain cells must be addressed. Strategies exist for larger molecules to enter the brain, including nanoparticle delivery systems [62], carrier-mediated transport [63], receptor-mediated transcytosis [64], or intranasal delivery [65] can be considered. Surface modifications or conjugation with specific ligands may also improve the transport properties of F1/F3. These aspects will be investigated in our future studies.

## Conclusions

This study indicates that F1/F3 peptides effectively inhibit the proliferation of U87 and U118 glioblastoma cells. The observed changes in mitochondrial function, cell cycle progression, and activation of stress response pathways provide insights into the potential mechanisms underlying the anti-cancer effects of these peptides. Further studies may delve into elucidating the specific pathways involved and exploring the translational potential of F1/F3 as anti-glioblastoma therapeutics.

## Supporting information

**S1 Fig. The comparison of cell cycles of glioblastoma cells with and without F1/F3 treatment.** The concentration of F1/F3 was 5 and 10 µg/ml in the treatment of U87 and U118 cells, respectively.
(TIF)

**S2 Fig. GSEA of untreated U87 and U118 cells.** (**A**) GSEA of the Hallmark pathways in the U87 relative to the U118 groups. Gene ontology terms enriched in between the U87F and U118 groups: (**B**) biological process; (**C**) molecular function; and (**D**) cellular component.
(TIF)

**S3 Fig. Gene ontology enrichment comparison between U87F and U87 groups.** Top 20 molecular function terms (**A**) and cellular component terms (**B**) enriched in the U87F group with respect to the U87 group.
(TIF)

**S4 Fig. Western blotting analysis of JUNB and PZP in the U87F group, and glycolysis/ OXPHOS assay for U118 cells.** (**A**) and PZP (**B**) regulation in the U87 cells treated with different concentrations of F1/F3. (**C**) Evaluation of glycolysis/OXPHOS levels in the U118 (top) and U118F (bottom) groups. **: $P$-value < 0.01, ***: $P$-value < 0.001, and ****: $P$-value < 0.0001.
(TIF)

**S1 Raw images. Raw western blot images used in Fig 7 and S4 Fig.**
(PDF)

**S1 Table. Quantitative proteomic comparison of the protein profiles between different groups.**
(XLSX)

**S2 Table. Supporting peptides identified by LC-MS/MS in the proteomic analysis.**
(XLSX)

**S3 Table. The analysis results of western-blot and TCGA data.**
(XLSX)

**S4 Table. Glycolysis/OXPHOS test data.**
(XLSX)

## Acknowledgments

We thank Professor Abigail Elizur for her valuable advice and support.

## Author Contributions

**Conceptualization:** Xiaosong Liu, Tianfang Wang.

**Data curation:** Yichen Wang.

**Formal analysis:** Yichen Wang, Fengyun Xiao, Xiaosong Liu, Guoying Ni.

**Funding acquisition:** Xiaosong Liu, Guoying Ni, Tianfang Wang.

**Investigation:** Yichen Wang, Furong Zhong, Fengyun Xiao, Xiaosong Liu.

**Methodology:** Junjie Li, Xiaosong Liu.

**Project administration:** Wei Zhang.

**Resources:** Xiaosong Liu, Wei Zhang.

**Software:** Junjie Li, Tianfang Wang.

**Supervision:** Xiaosong Liu, Guoying Ni, Wei Zhang.

**Validation:** Yichen Wang, Furong Zhong, Fengyun Xiao.

**Visualization:** Yichen Wang, Tianfang Wang.

**Writing – original draft:** Yichen Wang, Xiaosong Liu, Tianfang Wang.

**Writing – review & editing:** Xiaosong Liu, Guoying Ni, Tianfang Wang.

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
