## [Decision Letter · Decision Letter 0]

2 Apr 2024

PONE-D-24-09044Host-defence caerin 1.1 and 1.9 peptides suppress glioblastoma U87 and U118 cell proliferation through the modulation of mitochondrial respiration and induce the downregulation of CHI3L1PLOS ONE

Dear Dr. Wang,

Thank you for submitting your manuscript to PLOS ONE. After careful consideration, we feel that it has merit but does not fully meet PLOS ONE’s publication criteria as it currently stands. Therefore, we invite you to submit a revised version of the manuscript that addresses the points raised during the review process.

We look forward to receiving your revised manuscript.

Kind regards,

Jianxun Ding, Ph.D.

Academic Editor

PLOS ONE

Journal Requirements:

"This study was supported in part by the First Affiliated Hospital of Guangdong Pharmaceutical University, Deng Feng project of Foshan First People’s Hospital (2019A008), Foshan municipal Government (2015AG1003), Guangdong Science and Technology Department (2016A020213001), National Science Foundation of Guangdong province (2020A1515010855), National Natural Science Foundation of China (31971355). The funders had no role in study design, data collection and analysis, decision to publish, or preparation of the manuscript."

Reviewers' comments:

Reviewer's Responses to Questions

**Comments to the Author**

1. Is the manuscript technically sound, and do the data support the conclusions?

Reviewer #1: Yes

Reviewer #2: Yes

2. Has the statistical analysis been performed appropriately and rigorously? 

Reviewer #1: N/A

Reviewer #2: I Don't Know

3. Have the authors made all data underlying the findings in their manuscript fully available?

Reviewer #1: Yes

Reviewer #2: Yes

4. Is the manuscript presented in an intelligible fashion and written in standard English?

Reviewer #1: Yes

Reviewer #2: Yes

5. Review Comments to the Author

Reviewer #1: 1) Is the 95% purity of the synthesied peptides enough? Maybe more than 95% is better? If there is possibility to synthesize more than 95%, this should me mentioned or the reason for 95% should me mentioned.

2) Assays and analyses were properly planned and designed.

3) Electron microscopy images are missing and they were not presented and commented throughout the text.

4) F1 and F3 induced/activated mechanisms and effects were discussed in an understandable and sufficient detailed way.

5) In line and contraversary findings and aspects were discussed.

6) Possible activity mechanisms were also discussed well in the discussion section.

Reviewer #2: The manuscript (PONE-D-24-09044_reviewer)

Host-defence caerin 1.1 and 1.9 peptides suppress glioblastoma U87 and U118 cell proliferation through the modulation of mitochondrial respiration and induce the downregulation of CHI3L1

The manuscript demonstrates ability of certain peptides (F1/F3) to reduce glioblastoma cell lines proliferation, and migration while increasing apoptosis. The authors performed deep proteomic analysis and identified CHI3L1 which separate GBM patient’s survival based on its expression. The study is very good but need major correction to be suitable for publication.

The study has Major issues:

• In the description about used cell lines authors have to mention all major mutations relevant to the cell lines.

• Under 2.3 CCK8 Assay, authors did not explain the time between cultivation and treatment with peptide. Authors should give more detailed protocol. How long were U97 and U118 cultivated in 96 well plate before introduction of peptides?

• Under 2.5 Fluorescence microscopy, authors should elaborate more about F1/F3 and P3 marked by FITC. How is that? Are they linked with a fluorophore, or antibody or else (explain)? Is it covalent link, non-covalent or else?

• Authors has mentioned “Blank group” multiple times. It is more appropriate to write “untreated group”.

• Under 2.6 Cell apoptosis assay, authors should define the number of cells seeded in the 6 well plates before adding F1, F3 or P3. In addition, authors should define the incubation time before treatment.

• Under 2.7 Cell cycle experiment, authors should give more details to the protocol. Information needed: Number of cells seeded in the 6 well plate, incubation time, treatment concentration.

• Under 2.8 Electron microscope photography, authors should mention the treatment concentration used for F1, F3 and P3.

• Page 8 line 167, authors wrote “electron microscope (FACSAria II; BD Biosciences, San Jose, CA, USA)” FACSAria is not an electron microscop. Correction needed.

• In Lines, 173, 182, authors mentioned “tissue” However in the manuscript authors only used cell lines. Correction needed. It is “cells”

• Lines 204 – 205, the link to the data set “PXD044941” mut be provided.

• Line 236 – 238, Authors should explain why Caerin peptide was (range from 0 to 10 μg/mL) for U87 but (range from 0 to 20 μg/mL) for U118. Why they are not the same?

• Line 250, authors should mention how long was exposure to peptide (concentration of 5 μg/mL)?

• Authors mentioned using a combination of F1 and F3. However, they did not mention the concentration of each compound and the percentage used after combination if applicable. Authors should provide these information for each assay.

• Line 277, authors defined IC50 as 5.018 μg/ml. However, authors did not mention how much represent F1 and how much of it is F3. This information is important.

• Fig 1C, need better resolution.

• In the discussion , discuss a possible reason that CHI3L1 did not respond to F1/F3 in a dose dependent matter on U118 while the opposite for u87.

• It was not mentioned how many times assays were performed independently, is it 1, 2 or 3?

• For proteomic analysis authors should explain whether cell lines were used in at least 2 independent analysis or not.

The study has some minor issues:

• In Fig 3, it is better to rearrange images to show FITC-green separate, DAPI-blue separate and merge separate.

6. PLOS authors have the option to publish the peer review history of their article (what does this mean?). If published, this will include your full peer review and any attached files.

Reviewer #1: No

Reviewer #2: **Yes: **Dr. Bahauddeen Alrfaei

---

## [Author Response · Author response to Decision Letter 0]

19 Apr 2024

Dear editor and reviewers,

Thank you very much for editing and reviewing our manuscript. Please find our point-by-point answers to the questions raised by the reviewers. Changes to the manuscript are shown in red font in the "'Revised Manuscript with Track Changes".

Editor comments:

RESPONSE: We have formatted the manuscript to meet the style requirements (shown in red font in the revised manuscript) and named the files accordingly.

2. Please provide an amended statement that declares *all* the funding or sources of support (whether external or internal to your organization) received during this study. Please also include the statement “There was no additional external funding received for this study.” in your updated Funding Statement. Please include your amended Funding Statement within your cover letter.

RESPONSE: Corrected and the updated Funding Statement has been added to the cover letter.

3. PLOS ONE now requires that authors provide the original uncropped and unadjusted images underlying all blot or gel results reported in a submission’s figures or Supporting Information files.

RESPONSE: The raw images were included in S1 File in the original submission. This has been renamed as S5 Fig in the revised manuscript, with the loading order, identity of experimental samples clearly labelled. The method used to capture the image has been added.

Reviewer #1

1. Is the 95% purity of the synthesied peptides enough? Maybe more than 95% is better? If there is possibility to synthesize more than 95%, this should me mentioned or the reason for 95% should me mentioned.

RESPONSE: We have checked the documents provided by Qiangyao Biological Technology Co., Ltd. The purity of F1, F3 and P3 is 99.47%, 99.55% and 99.29%, respectively. These have been added in the revised manuscript.

2. Assays and analyses were properly planned and designed.

RESPONSE: We are grateful that the reviewer acknowledges the planning and design of the assays and analyses.

3. Electron microscopy images are missing and they were not presented and commented throughout the text.

RESPONSE: We are uncertain about which electron microscopy images the reviewer is referring to. The electron microscopy images are presented in Fig 4 and discussed in the paragraph preceding Fig 4.

4. F1 and F3 induced/activated mechanisms and effects were discussed in an understandable and sufficient detailed way.

RESPONSE: We are grateful that the reviewer found our discussion on F1/F3 associated mechanisms to be clear and comprehensive.

5. In line and contraversary findings and aspects were discussed.

RESPONSE: We appreciate the reviewer's acknowledgment that our discussion covers both congruent and contentious findings.

6. Possible activity mechanisms were also discussed well in the discussion section.

RESPONSE: We are grateful for the reviewer's recognition of our efforts in the thorough discussion on possible activity mechanisms in the discussion section.

Reviewer #2

1. In the description about used cell lines authors have to mention all major mutations relevant to the cell lines.

RESPONSE: The mutations relevant to the cell lines have been added to the Materials and Methods. 

2. Under 2.3 CCK8 Assay, authors did not explain the time between cultivation and treatment with peptide. Authors should give more detailed protocol. How long were U97 and U118 cultivated in 96 well plate before introduction of peptides?

RESPONSE: Before the addition of the peptides, U87 and U118 cells were cultivated in flat-bottomed 96-well plates for 18 h to ensure successful cell adhesion. This information has been added to the Materials and Methods.

3. Under 2.5 Fluorescence microscopy, authors should elaborate more about F1/F3 and P3 marked by FITC. How is that? Are they linked with a fluorophore, or antibody or else (explain)? Is it covalent link, non-covalent or else?

RESPONSE: The connection between the peptides and FITC is achieved by the reaction between the isothiocyanate group of FITC and the amino group of the peptide. This information has been added to the Materials and Methods.

4. Authors has mentioned “Blank group” multiple times. It is more appropriate to write “untreated group”.

RESPONSE: “Blank group” has been changed to “untreated group”.

5. Under 2.6 Cell apoptosis assay, authors should define the number of cells seeded in the 6 well plates before adding F1, F3 or P3. In addition, authors should define the incubation time before treatment.

RESPONSE: U87 and U118 cells were cultured for 18 hours in two flat bottomed 6-well plates, with 5×105 cells seeded before adding the peptides. This information has been added.

6. Under 2.7 Cell cycle experiment, authors should give more details to the protocol. Information needed: Number of cells seeded in the 6 well plate, incubation time, treatment concentration.

RESPONSE: U87 and U118 cells were cultured for 18 hours in two flat bottomed 6-well plates, with 5×105 cells seeded before adding the peptides. This information has been added.

7. Under 2.8 Electron microscope photography, authors should mention the treatment concentration used for F1, F3 and P3.

RESPONSE: The concentration is 5 μg/mL and has been added.

8. Page 8 line 167, authors wrote “electron microscope (FACSAria II; BD Biosciences, San Jose, CA, USA)” FACSAria is not an electron microscop. Correction needed.

RESPONSE: We apologise for the oversight, which should have been avoided. The correct model of the electron microscope is the Hitachi 7500 (HITACHI, Japan). Thank you for bringing this to our attention, and the error has been rectified.

9. In Lines, 173, 182, authors mentioned “tissue” However in the manuscript authors only used cell lines. Correction needed. It is “cells”

RESPONSE: Corrected.

10. Lines 204 – 205, the link to the data set “PXD044941” mut be provided.

RESPONSE: While the manuscript is still under review, the dataset remains private. However, reviewers have access to it through the link (https://www.ebi.ac.uk/pride/) using the credentials provided in the cover letter: reviewer_pxd044941@ebi.ac.uk, password: x9EzwCNx. Upon acceptance of the manuscript, the dataset will be made publicly available. Readers can access it using the same link and searching for "PXD044941".

11. Line 236 – 238, Authors should explain why Caerin peptide was (range from 0 to 10 μg/mL) for U87 but (range from 0 to 20 μg/mL) for U118. Why they are not the same?

RESPONSE: The concentration gradient was designed based on their respect IC50 values, which were 5.018 and 11.180 μg/mL. The explanation has been included in the revised manuscript.

12. Line 250, authors should mention how long was exposure to peptide (concentration of 5 μg/mL)?

RESPONSE: The exposure was 18 hr and the concentration was 5 μg/mL. The information has been added. 

13. Authors mentioned using a combination of F1 and F3. However, they did not mention the concentration of each compound and the percentage used after combination if applicable. Authors should provide these information for each assay.

RESPONSE: The molar ratio of F1 and F3 was 1:1, and the concentration used was 5 μg/mL. This information has been added accordingly.

14. Line 277, authors defined IC50 as 5.018 μg/ml. However, authors did not mention how much represent F1 and how much of it is F3. This information is important.

RESPONSE: The molar ration of F1 to F3 was 1:1. This information has been added to the “CCK8 Assay” section of Materials and Methods.

15. Fig 1C, need better resolution.

RESPONSE: The resolution has been improved.

16. In the discussion, discuss a possible reason that CHI3L1 did not respond to F1/F3 in a dose dependent matter on U118 while the opposite for u87.

RESPONSE: Possible reasons have been discussed in the revised manuscript. 

17. It was not mentioned how many times assays were performed independently, is it 1, 2 or 3?

RESPONSE: The data are representative of at least 3 independent experiments, and this information has been added.

18. For proteomic analysis authors should explain whether cell lines were used in at least 2 independent analysis or not.

RESPONSE: Cell lines were used in three independent analysis. This information has been added in figure legend.

19. In Fig 3, it is better to rearrange images to show FITC-green separate, DAPI-blue separate and merge separate.

RESPONSE: The FITC-green, DAPI-blue and merge have been separately shown in revised Fig 3.

The responses provided above address the comments from the editor and the reviewers. We would like to express our gratitude for your valuable input and time in reviewing our work.

Sincerely

Tianfang Wang, Ph.D.

---

## [Decision Letter · Decision Letter 1]

30 Apr 2024

PONE-D-24-09044R1Host-defence caerin 1.1 and 1.9 peptides suppress glioblastoma U87 and U118 cell proliferation through the modulation of mitochondrial respiration and induce the downregulation of CHI3L1PLOS ONE

Dear Dr. Wang,

Thank you for submitting your manuscript to PLOS ONE. After careful consideration, we feel that it has merit but does not fully meet PLOS ONE’s publication criteria as it currently stands. Therefore, we invite you to submit a revised version of the manuscript that addresses the points raised during the review process.

It would be better to insert scales into Fig. 4 so the reader could get idea on the scales of the system under study.

We look forward to receiving your revised manuscript.

Kind regards,

Jianxun Ding, Ph.D.

Academic Editor

PLOS ONE

Journal Requirements:

Additional Editor Comments:

It would be better to insert scales into Fig. 4 so the reader could get idea on the scales of the system under study.

Reviewers' comments:

Reviewer's Responses to Questions

**Comments to the Author**

1. If the authors have adequately addressed your comments raised in a previous round of review and you feel that this manuscript is now acceptable for publication, you may indicate that here to bypass the “Comments to the Author” section, enter your conflict of interest statement in the “Confidential to Editor” section, and submit your "Accept" recommendation.

Reviewer #1: All comments have been addressed

Reviewer #2: All comments have been addressed

2. Is the manuscript technically sound, and do the data support the conclusions?

Reviewer #1: Yes

Reviewer #2: Yes

3. Has the statistical analysis been performed appropriately and rigorously? 

Reviewer #1: N/A

Reviewer #2: N/A

4. Have the authors made all data underlying the findings in their manuscript fully available?

Reviewer #1: Yes

Reviewer #2: Yes

5. Is the manuscript presented in an intelligible fashion and written in standard English?

Reviewer #1: Yes

Reviewer #2: Yes

6. Review Comments to the Author

Reviewer #1: It would be better to insert scales into Fig. 4 so the reader could get idea on the scales of the system under study.

Reviewer #2: (No Response)

7. PLOS authors have the option to publish the peer review history of their article (what does this mean?). If published, this will include your full peer review and any attached files.

Reviewer #1: **Yes: **Ozan Unsalan

Reviewer #2: **Yes: **Bahauddeen Alrfaei

---

## [Author Response · Author response to Decision Letter 1]

1 May 2024

Dear editor and reviewers,

Thank you very much for editing and reviewing our manuscript. Please find our answer to the question raised by the reviewer. Changes to the manuscript are shown in red font in the "'Revised Manuscript with Track Changes".

Reviewer #1

1. It would be better to insert scales into Fig. 4 so the reader could get idea on the scales of the system under study.

RESPONSE: We have added the scale bar of the system under study to revised Fig 4 and one sentence in the figure legend.

The response provided above addresses the comment from Reviewer 1. We would like to express our gratitude for your valuable input and time in reviewing our work.

Sincerely

Tianfang Wang, Ph.D.

---

## [Decision Letter · Decision Letter 2]

8 May 2024

Host-defence caerin 1.1 and 1.9 peptides suppress glioblastoma U87 and U118 cell proliferation through the modulation of mitochondrial respiration and induce the downregulation of CHI3L1

PONE-D-24-09044R2

Dear Dr. Wang,

We’re pleased to inform you that your manuscript has been judged scientifically suitable for publication and will be formally accepted for publication once it meets all outstanding technical requirements.

Kind regards,

Jianxun Ding, Ph.D.

Academic Editor

PLOS ONE

Additional Editor Comments (optional):

The revised manuscript is ready for publication.

Reviewers' comments:

Reviewer's Responses to Questions

**Comments to the Author**

1. If the authors have adequately addressed your comments raised in a previous round of review and you feel that this manuscript is now acceptable for publication, you may indicate that here to bypass the “Comments to the Author” section, enter your conflict of interest statement in the “Confidential to Editor” section, and submit your "Accept" recommendation.

Reviewer #1: All comments have been addressed

2. Is the manuscript technically sound, and do the data support the conclusions?

Reviewer #1: Yes

3. Has the statistical analysis been performed appropriately and rigorously? 

Reviewer #1: Yes

4. Have the authors made all data underlying the findings in their manuscript fully available?

Reviewer #1: Yes

5. Is the manuscript presented in an intelligible fashion and written in standard English?

Reviewer #1: Yes

6. Review Comments to the Author

Reviewer #1: Dear authors,

All of my questions have been addressed well by yourselves. This study is of high importance in the field, in my opinion and congratulations. Best regards.

7. PLOS authors have the option to publish the peer review history of their article (what does this mean?). If published, this will include your full peer review and any attached files.

Reviewer #1: **Yes: **Ozan Unsalan

---

## [Editor Report · Acceptance letter]

14 May 2024

PONE-D-24-09044R2 

PLOS ONE

Dear Dr. Wang, 

I'm pleased to inform you that your manuscript has been deemed suitable for publication in PLOS ONE. Congratulations! Your manuscript is now being handed over to our production team.

Kind regards, 

on behalf of

Dr. Jianxun Ding 

Academic Editor

PLOS ONE